# Spatial Organisation and Invasive Behaviour of Metastatic Cutaneous Squamous Cell Carcinoma-Derived Multicellular Spheroids Reflect Tumour Cell Phenotype

**DOI:** 10.3390/cancers17213399

**Published:** 2025-10-22

**Authors:** Benjamin Genenger, Jessica Conley, Chelsea Penney, Luke McAlary, Jay R. Perry, Bruce Ashford, Marie Ranson

**Affiliations:** 1School of Science and Molecular Horizons, Faculty of Science, Medicine and Health, University of Wollongong, Wollongong, NSW 2522, Australia; auy7wq@virginia.edu (B.G.); cpenney@uow.edu.au (C.P.); lmcalary@uow.edu.au (L.M.); jayp@uow.edu.au (J.R.P.); 2Graduate School of Medicine, University of Wollongong, Wollongong, NSW 2522, Australia; bruceash@uow.edu.au; 3Illawarra Shoalhaven Local Health District, NSW Health, Wollongong, NSW 2500, Australia

**Keywords:** cutaneous squamous cell carcinoma, cSCC, in toto spheroid imaging, microscopy, multicellular tumour spheroids, MCTS, epithelial–mesenchymal transition, EMT, live-cell imaging, invasion

## Abstract

**Simple Summary:**

Cutaneous squamous cell carcinoma (cSCC) is a common skin cancer, especially in the head and neck, and can spread to nearby lymph nodes. Two key players in how this cancer grows and spreads are certain cancer cell changes (called EMT—epithelial–mesenchymal transition) and support cells known as fibroblasts. In this study, we recreated three-dimensional clusters of cancer and fibroblast cells called multicellular tumour spheroids, using samples derived from the lymph nodes of patients with metastatic cSCC. We analysed how these cell clusters formed, how they were structured, and how they invaded surrounding tissue. We found that the way the cancer cells behaved and invaded was linked to their EMT status. These multicellular tumour spheroid models help us to better understand how cSCC spreads and could lead to improved ways to study and treat this aggressive form of skin cancer.

**Abstract:**

**Background/Objectives**: Cutaneous squamous cell carcinoma (cSCC) is a very common skin malignancy of the head and neck area, with a propensity to spread to local lymph nodes. Epithelial-to-mesenchymal transition (EMT) and cancer-associated fibroblasts (CAFs) play a well-documented role in the progression of the disease. In this study, we developed and characterised multicellular tumour spheroids (MCTS) composed of patient-derived metastatic cSCC cell lines—each exhibiting distinct phenotypes—combined with either dermal- or lymph node-derived fibroblasts. We aimed to investigate how these cellular combinations influence MCTS formation, spatial architecture, and invasive behaviour. We hypothesised that the interplay between different cSCC and fibroblast cell combinations would differentially influence spheroid formation and invasion. **Methods**: Using live-cell microscopy we assessed the spatial architectures specific to each cell line-fibroblast combination and evaluated the expression of EMT and CAF markers. Furthermore, we utilised MCTS in invasion models to investigate associations between the mode of invasion and the EMT phenotype of the cancer cell line. **Results**: We show that metastatic cSCC/fibroblast MCTS self-organise into distinct spatial architectures. They also invade through collagen in a manner influenced by fibroblasts but dominated by the EMT status of the originating cancer cells. **Conclusions**: These findings highlight the physiological relevance and utility of MCTS as models for investigating tumour–stroma interactions and invasion dynamics in metastatic cSCC.

## 1. Introduction

Cutaneous squamous cell carcinoma (cSCC) is a very common cancer of the keratin-producing cells of the skin. Annual age-adjusted primary incidence rates for cSCC are as high as 856/100,000 in coastal New South Wales (Australia) [1]. The majority of cSCC (>95%) arises in the sun-exposed skin of the head and neck (HN-cSCC), driven by ultraviolet radiation exposure with other contributing risk factors including age, sex, skin type, and immunosuppression status [2]. Advanced HN-cSCC presents with metastatic spread in up to 5% of cases. Metastatic HN-cSCC frequently spreads from the primary site to the cervical and parotid lymph nodes, with distant organ metastasis being less common (<6% of metastatic cSCC) [3,4,5,6].

Epithelial-to-mesenchymal transition (EMT) is the dynamic process whereby epithelial cells lose their adhesive properties and acquire mesenchymal features associated with a more invasive phenotype [7,8]. EMT plays an important role during the initial acquisition of invasive properties and disease progression of cSCC, as recently reviewed by Genenger et al. [9]. Interestingly, multiple studies report an association of EMT phenotype and cancer-associated fibroblasts (CAFs), a stromal cell type comprising a large proportion of the tumour [10,11,12]. In primary cSCC, Ji et al. found an enrichment of CAFs and cancer cells exhibiting an EMT phenotype in a fibrovascular niche along the leading edge of tumours [13]. Characterisation of ligand–receptor interactions identified TGF-β as a central regulator of CAF activation. This was in line with previous findings supporting a role of TGF-β-activated CAFs in inducing EMT and facilitating invasion in cSCC [14].

Some research suggests that the subtype of fibroblast giving rise to the CAFs differentially influences EMT and invasion. At the primary site of invasion (i.e., the skin), the most abundant subtypes are reticular and papillary dermal fibroblasts (DFs). These subtypes reportedly have distinct functions in healthy skin, exerted through their differential expression of extracellular matrix (ECM) components, growth factors, and proteases [15,16,17]. In cSCC, there is evidence that papillary dermal fibroblasts can limit cancer cell invasion, while reticular dermal fibroblasts are tumour-promoting and predisposed to acquire an activated CAF phenotype [18]. Using organotypic skin models, Hogervorst et al. demonstrated that cSCC cell lines of different progression stages invade more extensively into a dermis containing reticular fibroblasts compared to papillary fibroblasts [18]. In this reticular fibroblast/cSCC co-culture model, reticular fibroblasts expressed CAF biomarkers α-smooth muscle actin (α-SMA), and vimentin. Conversely, cSCC cells expressed EMT markers N-cadherin and ZEB1.

However, how lymph node-derived fibroblasts (LNFs) compared to dermally derived fibroblasts (DFs) shape the metastatic niche in the affected lymph nodes, their influence on EMT, and their expression of CAF markers in the context of metastatic cSCC remains unclear. Given the distinct tumour microenvironment of the lymph node compared to the skin, alternative models to skin equivalents are necessary for this determination.

In pursuit of increased physiological relevance over single-cell monolayers and spheroids, stromal cells and cancer cells have been combined in multicellular tumour spheroids (MCTS) [19,20]. MCTS are self-assembling three-dimensional (3D) spheroids containing multiple distinct cell types to replicate the cellular composition of the microenvironment. Like conventional single-cell spheroids, MCTS recapitulate nutrient, oxygen, and drug gradients, as well as homotypic cell–cell interactions [21,22]. However, MCTS also account for paracrine and heterotypic juxtacrine signalling effects between the cell types incorporated and therefore represent a more relevant in vitro model to study these effects on tumour cell biology [23,24]. For example, MCTS combining stromal and tumour cells can be useful tools for evaluating EMT, tumour–stroma interactions, and invasion.

In this study, we establish MCTS models of patient-derived cSCC cell lines (UW-CSCC1 and UW-CSCC2) and characterise the effects of the different cell combination on the formation and spatial architecture of MCTS. We hypothesised that the EMT status of the cancer cell and the origin of the fibroblast used would differentially influence the MCTS phenotype. Additionally, we examine the effect of dermal/lymph node fibroblasts on EMT- and CAF-associated immunofluorescent markers in MCTS invading into a collagen matrix to evaluate the effect of cancer and fibroblast cell phenotype on EMT and invasion in metastatic cSCC.

## 2. Materials and Methods

### 2.1. General Cell Culture and Maintenance

The patient-derived cell lines from lymph node cSCC metastases of the head and neck, UW-CSCC1 and UW-CSCC2, were used for the generation of MCTS and routinely cultured as previously described [25]. These cell lines have been extensively characterised and show distinctive genomic and phenotypic characteristics in vitro, reflective of their tumour of origin as well as their inter-tumour heterogeneity [25]. Dermal fibroblasts (DFs) are skin-derived telomerase-immortalised fibroblasts (of unknown papillary or reticular origin) and were obtained from Prof. P. Timpson (Garvan Institute, Sydney, Australia) [26]. The detailed composition of the cell culture media for single- and mixed-cell culture is included in Appendix A. Routine cell culture was conducted under low-oxygen conditions (3% O_2_, 5% CO_2_, 37 °C) unless otherwise indicated. Cell lines were routinely monitored for absence of mycoplasma contamination.

### 2.2. Generation of Patient-Derived Lymph Node Fibroblasts

Patient-derived lymph node fibroblasts (LNFs) were cultured from uninvolved lymphatic tissue obtained after a radical neck dissection (as previously published by Perry et al. [25]). Patient tissue was collected in accordance with the Declaration of Helsinki and University of Wollongong Human Research Ethics Committee’s approval (HE14/397). The tissue was dissected into 1 mm^3^ cubes within 1 h of the surgery and the explants plated onto a tissue culture surface in LNF cell culture medium (Appendix A). The lymphatic immune cell population was removed during biweekly media changes. Other potentially contaminating cells were removed via differential trypsinisation, based on the rapid detachment of fibroblasts. Cultures were incubated with 0.05% trypsin-EDTA for 30 s intervals until cells visibly lifted. Upon reaching confluency, the fibroblasts were detached and transferred to a new culture vessel. The identity of LNFs was confirmed based on their morphological characteristics, positive staining for vimentin, and a lack of cytokeratin (Appendix A). LNFs were routinely cultured in the same manner as DFs, detailed above.

### 2.3. Generation of Fluorescent Katushka2S Transfected UW-CSCC Cell Lines, UW-CSCC1-K and UW-CSCC2-K

A pcDNA3.1(+) plasmid containing Katushka2S under the control of a CMV promoter (Appendix A) was designed in-house and synthesised by Gene Universal (Newark, DE, USA). Katushka2S is a GFP-like far-red fluorescent protein with bright and rapidly maturing fluorescence, which is suitable for challenging samples such as MCTS [27]. Plasmid DNA was propagated in chemically competent DH5α cells and isolated using a HiSpeed^®^ Plasmid Maxi Kit (Qiagen, Hilden, Germany) as per manufacturer’s instructions. UW-CSCC1 and UW-CSCC2 cells were seeded in 12-well plates (100,000 cells/well) and transfected with Katushka2S plasmid DNA using Lipofectamine^TM^ 3000 as per manufacturer’s instructions (Invitrogen, Waltham, MA, USA). After 48 h, cells with stable genomic integration of the plasmid were selected for using Geneticin (1 mg/mL). Katushka2S-positive cells were enriched using three instances of fluorescence-assisted cell sorting (FACS) (Sony LE-MA900BP, Sony, Tokyo, Japan, λ_ex_ = 488 nm, Filter = 617/30: PE-Texas Red). Once 100% of the cells were Katushka2S positive, the cells were routinely cultured without Geneticin, and fluorescence was monitored by fluorescence microscopy (Appendix A). Transfected cells were denoted as UW-CSCC1-K and UW-CSCC2-K.

### 2.4. Immunocytochemistry of Two-Dimensional Cell Cultures

Immunocytochemistry of 2D cell cultures was conducted as previously published by Farrawell et al. [28]. Briefly, UW-CSCC cell lines were seeded in 8-well ibidi chamber slides (ibidi, Gräfelfing, Germany) at 20,000 cells/well. After 24 h, cells were fixed with 4% paraformaldehyde (PFA), permeabilised with Triton X-100, and non-specific binding sites were blocked with blocking buffer (10% foetal bovine serum (FBS), 2% bovine serum albumin (BSA), 0.1% Triton X-100). Primary vimentin (ab92547, Abcam, Cambridge, UK) (1:2000) and pan-cytokeratin (C2562, Sigma Aldrich, St. Louis, MO, USA) (1:2000) antibodies were incubated at 4 °C overnight. Fluorescent secondary antibody, anti-mouse AF488 (ab150105, Abcam, UK) (1:1000) was then incubated for 2 h at room temperature. The cells were counterstained with Hoechst-33342 (1:5000) and ActinRed^TM^ 555 (ThermoFisher, Waltham, MA, USA) (2 drops/mL). Stained slides were imaged using an oil-immersion objective (63×) on either an SP8 confocal microscope (Leica, Wetzlar, Germany) or DMi8 inverted microscope (Leica, Germany).

### 2.5. Generation and Immunocytochemistry of Multicellular Tumour Spheroids (MCTS)

MCTS were generated and processed as previously described by Genenger et al. [29]. Briefly, one cSCC cell line (UW-CSCC1 or UW-CSCC2) was combined with one fibroblast cell line (LNFs or DFs). For each cell type, 1500 cells per well were seeded in round-bottomed, ultra-low attachment 96-well plates (Costar^®^ Corning Incorporated, Corning, NY, USA). The cells were centrifuged at 300× *g* for 5 min to facilitate their aggregation. After six days of culture, ten MCTS per condition were collected and processed for imaging or resuspended in neutralised collagen I (1 mg/mL). Both collagen-embedded and non-embedded MCTS were transferred to ibidi chamber slides for fixation and staining. Non-embedded MCTS were left to adhere to the chamber bottom overnight and collagen-embedded MCTS were incubated under culturing conditions for 80 h prior to staining, as described previously [29].

MCTS were fixed with 4% PFA and permeabilised with Triton X-100. Permeabilised MCTS were stained for up to two proteins of interest, with ActinRed^TM^ 555 and Hoechst-33342 as counterstains for the actin cytoskeleton and nucleus, respectively. Dilutions and manufacturer details of antibodies are listed in Appendix A. Samples were cleared using 90:10 glycerol/tris (pH = 8.0) supplemented with the anti-fading agent N-propyl gallate (0.5% *w*/*v*) and sodium azide (0.02% *w*/*v*). All imaging was conducted using an SP8 FaLCon microscope (Leica, Germany) with a 20× multi-immersion objective with the adjustable correction collar set to glycerol (HC PL APO CORR CS2 20×/0.75 IMM (#11506343)).

### 2.6. Live-Cell Imaging of Multicellular Tumour Spheroid Formation and Collagen Invasion

For imaging of live-cell MCTS formation, LNFs and DFs were stained with CellTracker^TM^ Green CMFDA (Invitrogen, USA) as per the manufacturer’s instructions. Fluorescent UW-CSCC1-K or UW-CSCC2-K cells were seeded with the fluorescently tagged LNFs or DFs, as described above to form MCTS. Time-lapse imaging was conducted under normoxic temperature-controlled conditions (T = 37 °C) at 10× magnification (HC PL FLUOTAR 10×/0.32 DRY (#11506522)) on a Thunder DMi8 microscope (Leica, Germany). Z-stacks (z = 5) were acquired every 2.75 h for 60 h (23 time points). The timing and stack size represent a compromise between time resolution, spatial resolution, phototoxic effects, and photobleaching.

### 2.7. Imaging Processing, Analysis, and Quantification

Images were post-processed with Leica’s instant computational clearing (ICC). ICC maintains quantifiable fluorescence levels as it only removes background fluorescence levels without any deconvolution [30]. Maximum intensity projections of the final image in each time lapse were used for quantification (n = 6 per condition).

CellProfiler 4 [31] was used to quantify the fluorescence distribution (indicative of cell type distribution) in these MCTS. The inverted brightfield channel was used to determine the outline of the MCTS. The MCTS outlines were overlaid on fluorescent channels, and the fluorescence intensity was determined (module: MeasureObjectIntensityDistribution). Three adaptable bins (representing the core, intermediate zone, and shell of a spheroid) were used to determine the mean per pixel intensity levels for each cell type. The resulting data were analysed and graphed using GraphPad Prism v9.4.1. Statistical significance was determined via a two-way ANOVA with Tukey’s post hoc test for pairwise comparison.

### 2.8. MCTS Invasion Assay

MCTS were prepared as above and allowed to form for 48 h. Additionally, to evaluate baseline cancer cell invasion without fibroblast influence, spheroids of UW-CSCC1 or UW-CSCC2 alone were also prepared with the total number of cells equal to that of the MCTS. Spheroids and MCTS were transferred to ibidi chambers and embedded in neutralised collagen I (1 mg/mL) as above. Spheroids were allowed to invade for 0 h (control), 48 h (UW-CSCC1), or 18 h (UW-CSCC2). These time points were selected based on the known invasion propensities of these cell lines, as previously established [25]. Cells were then fixed, permeabilised, blocked, and stained for the actin cytoskeleton and nuclei, as above, and imaged using a DMi8 Thunder microscope (Leica, Germany). Invasion was quantified by determining the surface area of the spheroids in ImageJ v1.53. Invasion was calculated as a percentage increase in control surface area, and statistical analysis was performed using GraphPad Prism v9.4.1.

## 3. Results

### 3.1. Distinctive EMT Characteristics of Metastatic cSCC Cell Lines

Firstly, we determined the baseline expression levels of EMT markers in UW-CSCC1 and UW-CSCC2 to gauge the influence of the fibroblasts on their expression, utilising existing genomic and transcriptomic characterisations [25]. These data were used to predict the EMT phenotype of each cell line, which was then validated using immunofluorescent staining and Western blots (Appendix A), as summarised in Figure 1.

UW-CSCC1 is situated at the more mesenchymal end of the EMT spectrum. This is apparent in its elongated morphology and its higher expression of EMT transcription factors and the mesenchymal markers N-cadherin and vimentin compared to UW-CSCC2. Conversely, UW-CSCC2 shows more epithelial characteristics such as a cobblestone-like morphology and high levels of cytokeratin and the epithelial cell–cell adhesion protein E-cadherin (Appendix A; Appendix A). The more epithelial characteristics may be explained by the higher expression of OVOL-TFs (anti-EMT transcription factors) compared to UW-CSCC1 (Appendix A).

### 3.2. Metastatic cSCC and Fibroblast Combinations Drive Unique MCTS Spatial Architecture

To investigate the interplay between fibroblast types with these cell lines of distinctive EMT states, MCTS formation and organisational patterns were established using fluorescently tagged cell lines. Time-lapse imaging was employed to resolve any time-dependent effects (Appendix A). Co-cultures of DFs or LNFs and the cSCC cell lines readily formed MCTS within 2–3 days after seeding in ultra-low attachment plates at a 1:1 ratio (Figure 2). Distinctive architecture for each phenotypically distinct cell line combination was observed (Figure 2A,C). In samples containing LNFs, MCTS had an overall looser aggregation compared to DF MCTS, and often formed smaller “tandem spheroids”. LNF-containing spheroids were also slower to compact into a central spheroid mass, regardless of the tumour cell incorporated, as most evident 22 h post-seeding. Quantitative analysis of the cell type distribution confirmed the unique spatial composition of each cell line combination (Figure 2B,D). In UW-CSCC2 MCTS, there were significantly fewer cancer cells in the centre (Bin 1, B1, see also schematic in bottom left of Figure 2B,D) and intermediate zone (B2) of the spheroids compared to the spheroids’ shell (B3). The inverse was observed in the shell region, where the cancer cells were significantly more prominent. However, no significant differences in the distribution of cancer cells between UW-CSCC2 MCTS containing DFs or LNFs were observed. A fibroblastic core was less notable in UW-CSCC1 MCTS, with both cancer and fibroblast cells interspersed throughout the entire spheroid across both fibroblast subtype combinations. Interestingly, UW-CSCC1 and LNF spheroids assumed an intermediary phenotype with the beginnings of a fibroblastic core (significantly lower levels of cancer cells and higher levels of fibroblasts in B1 and B2 compared to UW-CSCC1/DF MCTS). These findings support the notion that both the fibroblast and cancer cell types used play a role in unique cell–cell interactions and driving the distinct spatial architectures of MCTS.

Notably, the spatial architecture formed by these MCTS resembles the spatial architecture observed in the original tumours (refer to Figure 1l,m in Perry et al. [25]). In the tumour from which UW-CSCC1 was derived, tumour cells were generally more interspersed with the surrounding stroma. Conversely, in the tumour from which UW-CSCC2 was derived, tumour cells grew in nests clearly separated from surrounding stroma, reflective of the spatial separation between fibroblast and cancer cells observed in UW-CSCC2 MCTS.

### 3.3. EMT Status of Metastatic cSCC Retained in Static and Invasive MCTS

To validate these findings and determine whether EMT states of cSCC cells are maintained or differentially influenced by fibroblast subtypes in MCTS, cell type and EMT marker distribution was confirmed by immunofluorescent staining (Figure 3). Metastatic cSCC cells were positively identified by staining for cancer cell-specific cytokeratin, a nucleus, and actin cytoskeleton. Fibroblasts were identified through the lack of cytokeratin but presence of nuclei and an actin cytoskeleton. UW-CSCC2 MCTS lacked cytokeratin-positive cells in the centre, whereas UW-CSCC1 did not, as expected (refer to Figure 2). The E-cadherin-positive UW-CSCC2 retained strong positive E-cadherin staining at cell–cell contact surfaces within the MCTS shell (Figure 3, Zoom-In 1 and 2) which was lacking from the actin-dense core of the MCTS, consistent with the fibroblastic core of live-cell MCTS containing UW-CSCC2 cells (Figure 2C,D). As observed above, the E-cadherin-negative UW-CSCC1 cells formed MCTS void of any clear organisational pattern, with cytokeratin-positive cancer cells interspersed with the fibroblasts (cytokeratin-negative cells). Collectively, these findings indicate that the positive and negative E-cadherin expression observed in 2D monoculture of UW-CSCC2 and UW-CSCC1, respectively, is conserved in MCTS co-culture models, and this likely influences the unique spatial architectures observed.

Interestingly, an additional new metastatic cSCC cell line UW-CSCC3—which appears to have an intermediate EMT phenotype and expresses moderate levels of E-cadherin compared to UW-CSCC1 and UW-CSCC2 (Appendix A)—forms an intermediate MCTS with DFs, with small fibroblast cores surrounded by loose cancer aggregates (Appendix A). As fluorescently tagged UW-CSCC3 cells were not available, live-cell spheroid formation imaging was not conducted on this cell line. However, imaging of stained UW-CSCC3/DF MCTS every 24 h for 3 days demonstrated minimal change in this loose, disorganised architecture over time. Together, this supports the idea of an E-cadherin-negative/mesenchymal and E-cadherin-positive/epithelial phenotype in UW-CSCC1 and UW-CSCC2, respectively, and that EMT states of the cSCC cell lines are retained in three-dimensional co-cultures with different fibroblast subtypes. Attempts to validate the role of E-cadherin in MCTS assembly by knockdown of E-cadherin in UW-CSCC2/DF MCTS using siRNA technology saw some alteration in MCTS architecture (Appendix A). Knockdown of approximately 50% of E-cadherin expression in UW-CSCC2 (similar to levels found in UW-CSCC3) resulted in a more disorganised, intermediate spheroid like the MCTS containing UW-CSCC3. Given the shared phenotype of UW-CSCC3/DF MCTS and UW-CSCC2-E-cadherin knockdown/DF MCTS—both with similarly moderate levels of E-cadherin—we can speculate that E-cadherin is likely driving the spheroid architecture in these models. However, transfection with a control siRNA also caused a change in MCTS phenotype, suggesting the transfection process itself is disruptive to UW-CSCC2, which could therefore interfere with functional interpretation of this E-cadherin knockdown.

### 3.4. Fibroblast Subtype Can Influence Invasion of Metastatic cSCC

Given that fibroblast subtypes are known to differentially influence the extent of cancer cell invasion, we next evaluated how DFs or LNFs impact invasion of metastatic cSCC cells into a simulated ECM. MCTS combinations were embedded into a neutralised collagen I matrix, and changes in spheroid surface area (relative to a time zero control) were measured as an indicator of invasion (Figure 4). All spheroids demonstrated an increase in surface area, consistent with invasion, regardless of the presence of fibroblasts or the subtype incorporated (Figure 4B,D). In general, MCTS containing DFs for both cell lines saw increased surface area, though this was only significant in UW-CSCC2, suggesting that the interaction between UW-CSCC2 and DFs was pro-invasive. It is worth noting that the invasion of UW-CSCC2/DF MCTS was over a shorter period than the UW-CSCC1/DF invasion, indicative of a rapid interaction between cell types to promote invasion. Conversely, despite the interspersed cell type distribution for UW-CSCC1 MCTS, the addition of either fibroblast subtype did not significantly impact spheroid invasiveness (Figure 4B).

Next, we investigated the expression of the mesenchymal phenotype marker vimentin in MCTS in the context of invasion. In collagen-embedded MCTS after 80 h of invasion (Figure 5), both cSCC cell lines maintained their expression of vimentin and cytokeratin compared to 2D culture (Figure 1) and non-embedded MCTS (Figure 3; Appendix A). UW-CSCC2 MCTS were completely dispersed, with cells spreading from the centre, and displayed high levels of cytokeratin but no or low levels of mesenchymal marker vimentin (i.e., CK+ and Vim−) (Figure 5; fibroblasts were Vim+ but there was no Vim+/CK+ colocalisation observed). UW-CSCC1 (CK+ and Vim+) cells invaded the collagen matrix as single cells in DF MCTS and as smaller groups in LNF MCTS, suggesting some influence of fibroblast subtype on UW-CSCC1 invasion. Staining for the CAF marker α-SMA and the ECM component fibronectin (FN1)—and thus the influence of cancer cell phenotype on the fibroblasts—was inconclusive (Appendix A). Overall, these results suggest that the mesenchymal and epithelial phenotypes displayed in the 2D culture are maintained in an invasive context.

## 4. Discussion

### 4.1. Epithelial cSCC Phenotype Correlates with Layered Architecture in MCTS

These investigations show how differential EMT phenotypes are influenced by cell–cell interactions and how this ultimately impacts the structure of co-culture models in vitro. We found a more epithelial cSCC phenotype (i.e., UW-CSCC2) was strongly associated with a more organised spatial architecture compared to those using a mesenchymal cSCC phenotype (i.e., UW-CSCC1) when combined in MCTS with fibroblasts derived from either the skin or lymph node. This likely reflects the cell–cell adhesion properties of these EMT states and their innate architectural propensities in three-dimensional culture, such as differential E-cadherin expression. E-cadherin is known to mediate spheroid aggregation and compaction [32], which could explain the E-cadherin-expressing UW-CSCC2 cell–cell interactions in a distinct layer separate from the fibroblasts. This formation of an epithelial shell that we observed has been previously reported; epithelial-like SW620 colon cancer cells enveloped a skin-derived fibroblast core in spheroids formed on a chitosan surface [33,34]. In contrast with our hypothesis regarding the influence of E-cadherin in driving this architecture, this previous study demonstrated a critical role of the fibroblasts’ N-cadherin expression [33]. However, when we consider that the fibroblasts did not form a compact core in the MCTS containing UW-CSCC1, there must be additional cell–cell dynamics underpinning the spheroid architecture we see here. For example, competing cell–cell adhesion properties of the cSCC cells and fibroblasts, in line with the differential adhesion hypothesis, are likely at play here [35,36].

### 4.2. Fibroblast Phenotype Can Influence Invasive Ability of Metastatic cSCC Cells

While distinctions in the way fibroblast subtypes influence cSCC cell invasion and EMT phenotypes have been elucidated between reticular and papillary skin fibroblasts [18], the role of lymph node fibroblasts on the tumour dynamics of cSCC metastases has not been previously demonstrated. We found that the invasion capacity of mesenchymal-like cSCC cells was not significantly altered between skin and lymph node fibroblast co-cultures, but the combination of dermal fibroblasts with the epithelial-like UW-CSCC2 cells significantly enhanced MCTS invasion into an ECM. This could be due to the fact that the LNFs were derived from uninvolved lymph nodes whereas the DFs are transformed fibroblasts and more likely to behave as CAFS, which are known help drive tumour cell EMT and invasiveness [9,10,12]. Interestingly, the EMT phenotype of both distinctive cSCC cell lines remained similar in all models, from 2D monocultures to invasive MCTS, suggesting that the fibroblast subtypes used here had little impact on the EMT behaviour of the cancer cells. This contrasts with previous co-culture modelling of cSCC with skin fibroblasts, where reticular dermal fibroblasts were demonstrated to increase the expression of EMT markers in cSCC cells to a greater extent than papillary dermal fibroblasts [18]. However, it is worth noting that this previous study utilised full-thickness skin equivalents to assess invasion dynamics. This highlights the importance of appropriate model selection for specific experimental questions. Such organotypic skin matrices or tumour-on-a-chip technologies can better recapitulate the larger anatomical context of a tumour such as ECM components (if not secreted by the cell types incorporated in MCTS), blood vessels, lymphatic vessels, and the skin-specific basement membrane [37]. While this is lacking in MCTS, these spheroid models remain ideal for the elucidation of close cell–cell contact interactions and their ability to recapitulate physiological gradients of oxygen, nutrients, and drugs in a scalable and reproducible manner [24].

### 4.3. Implications of MCTS Architecture and Invasion for Future Investigations

In this study, we investigated the spatial architecture of metastatic cSCC MCTS, as the structural organisation of MCTS has implications for their use as pre-clinical models. For example, a spatial organisation like in UW-CSCC2 MCTS would subject the cancer cells in the shell of the MCTS to higher doses of drugs, oxygen, and nutrients without establishing the desired gradients. The faulty assumption of equal cell type distribution in tumour MCTS models might introduce errors when looking at survival, stress markers, or necrosis markers to assess therapeutic efficacy in vitro [23]. Hence, careful characterisation of mixed-culture models and the cell type distribution of fibroblasts, immune cells, and cancer cells prior to usage in pre-clinical screenings such as that described here is advised.

This will be critical for the preclinical evaluation of treatments for metastatic cSCC specifically, given the phenotypic diversity of cell lines available for studying this disease and their likely effect on MCTS architecture as exemplified by this study. Importantly, the MCTS models presented in this study are highly versatile and can be easily altered to explore cancer cell interactions with other cell types (e.g., immune cells or endothelial cells) and the effects of therapeutic interventions such as immunotherapy.

## 5. Conclusions

In this study, we established and characterised MCTS derived from cSCC cell lines co-cultured with dermal and lymph node fibroblasts. MCTS self-organised to form unique spatial architectures for each cell line combination. Finally, we explored the use of MCTS in invasion models and propose an association of the mode of invasion with the EMT phenotype of the tumour cell line used. Future studies can expand upon the methodology presented here and could include immune cells to investigate aspects of anti-tumour immunity and the mediatory effect of different stromal cell types.

## Figures and Tables

**Figure 1 cancers-17-03399-f001:**
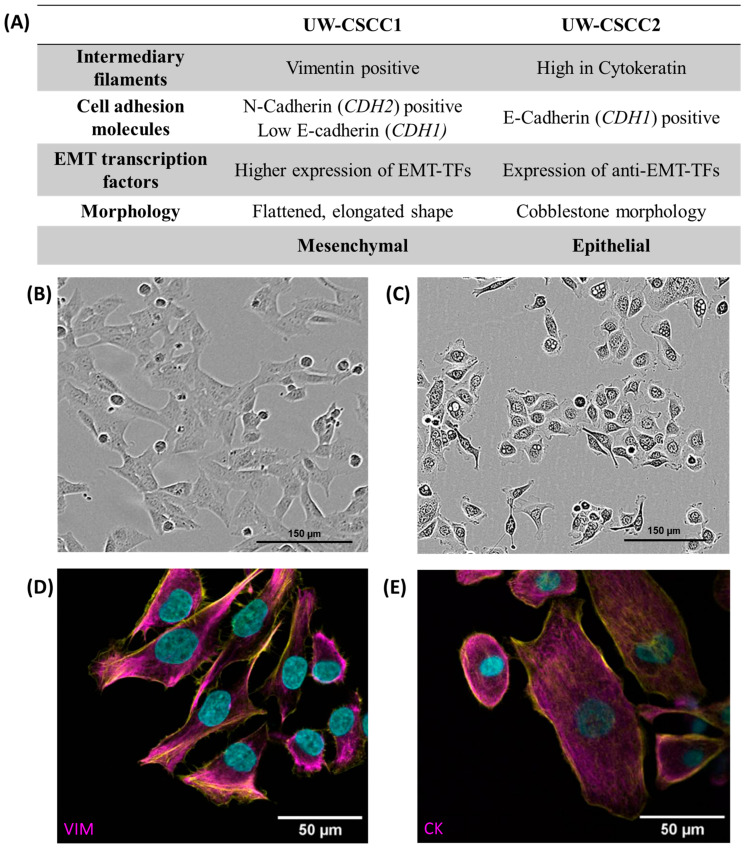
Baseline EMT characteristics of metastatic cSCC cell lines. Panel (**A**) contains a tabular overview of EMT characteristics of UW-CSCC cell lines cultured in standard 2D conditions, investigated by gene expression analysis and protein-based assays. Refer to Appendix A for transcription factor mRNA expression data, and Appendix A and Appendix A for EMT protein expression data. Panels (**B**,**C**) show the morphology of the cell lines UW-CSCC1 and UW-CSCC2, respectively, taken during routine culture. Panel (**D**) shows image of UW-CSCC1 stained for vimentin (magenta), and Panel (**E**) shows UW-CSCC2 stained for cytokeratin (magenta). Nuclei and actin cytoskeletons were counterstained in cyan and yellow, respectively. VIM: vimentin; CK: pan-cytokeratin. Length of scale bars indicated within each image.

**Figure 2 cancers-17-03399-f002:**
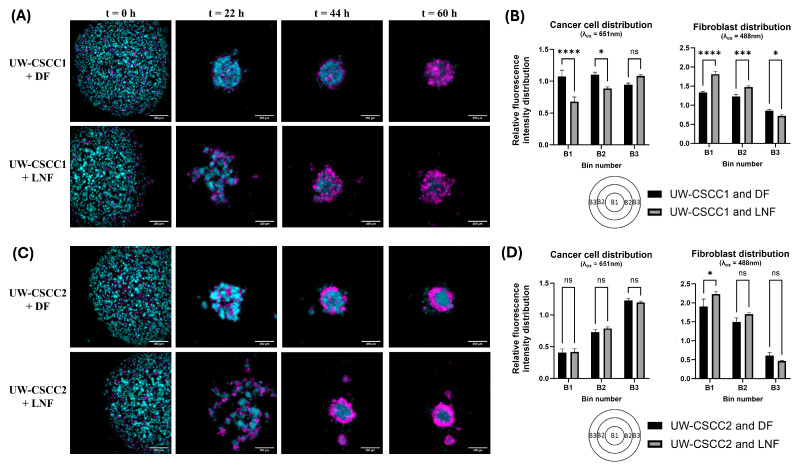
Differential spatial architecture of metastatic cSCC/fibroblast MCTS. Live fluorescently tagged fibroblasts (cyan) and patient-derived cSCC cell lines (magenta) were seeded into ultra-low attachment plates and imaged every 2.75 h for 60 h. (**A**,**C**) Representative images of spheroid spatial organisation. The differential architecture of the resulting spheroids shows distinct cell type distributions for UW-CSCC1 (Panel (**A**)) and UW-CSCC2 (Panel (**C**)). The images presented are ICC-cleared maximum-intensity projections (z = 5). The scale bar represents 250 µm. (**B**,**D**) Quantitative distribution of cell types in live fluorescently tagged cSCC/fibroblast MCTS. Quantitative analysis of the cancer cell (Panel (**B**,**D**), left) and fibroblast (Panel (**B**,**D**), right) distribution after 60 h spheroid formation shows statistically significant differences (*p* < 0.05; n = 6) of the mean fluorescence intensity in three zones (B1–B3: core, intermediate zone, shell) of the spheroids. Statistical significance was determined using a two-way ANOVA with Tukey’s post hoc test for pairwise comparisons (*, *p* < 0.05; ***, *p* < 0.001; ****, *p* < 0.0001). Not all statistically significant differences are displayed.

**Figure 3 cancers-17-03399-f003:**
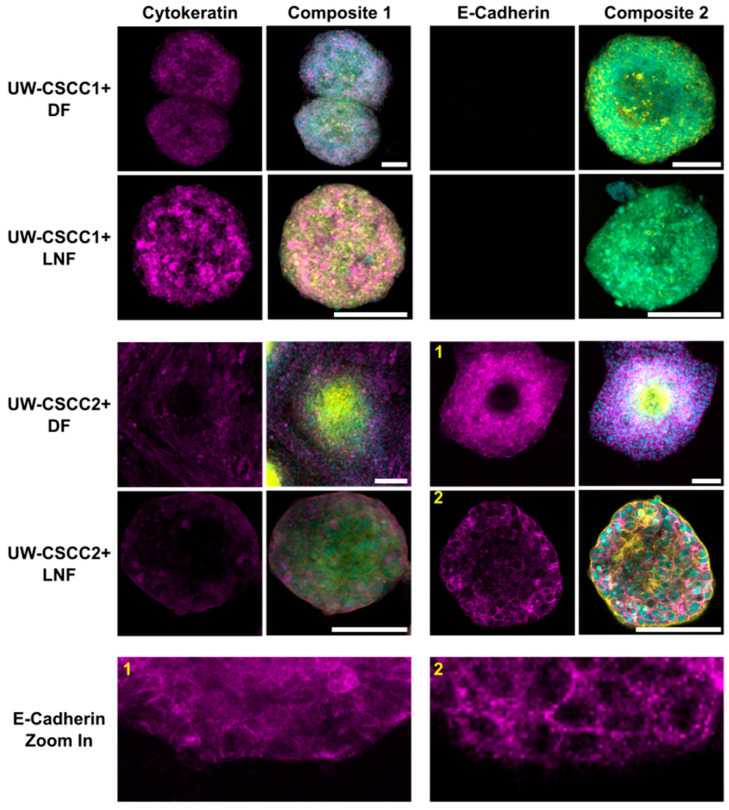
Epithelial markers cytokeratin and E-cadherin in metastatic cSCC MCTS. DFs or LNFs and UW-CSCC1 or UW-CSCC2 MCTS were formed for 6 days and stained for cytokeratin and E-cadherin (magenta). The nuclei (cyan) and actin cytoskeleton (yellow) were counterstained with Hoechst-33342 and ActinRed^TM^ 555, respectively, for both cytokeratin- (Composite 1) and E-cadherin (Composite 2)-stained spheroids. The images are each a singular z-slice through the centre of the spheroid. Scale bar represents 150 µm. E-cadherin Zoom in panel highlights detail of E-cadherin staining for (1) UW-CSCC2 + DF MCTS and (2) UW-CSCC2 + LNF MCTS (numbers in yellow denote the image from which the respective zoom-in image was taken).

**Figure 4 cancers-17-03399-f004:**
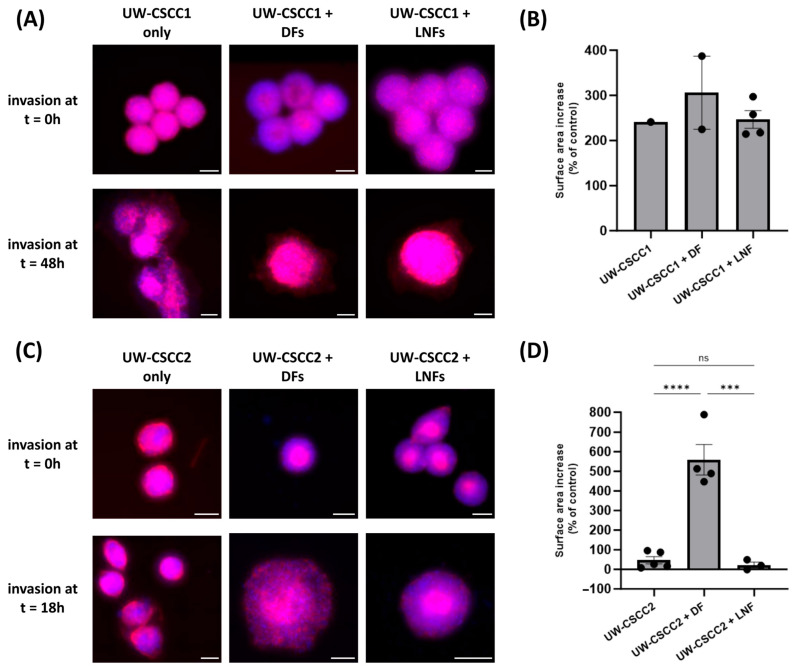
Invasion of metastatic cSCC single-cell spheroids and MCTS into a collagen I matrix. Representative UW-CSCC1 (Panel (**A**)) or UW-CSCC2 (Panel (**C**)) single-cell spheroids or MCTS with either DFs or LNFs were embedded into collagen I and their invasion was assessed. Scale bar represents 150 µm. Surface area of invaded spheroids containing UW-CSCC1 (Panel (**B**); time = 48 h) and UW-CSCC2 (Panel (**D**); time = 18 h) determined by nuclei (blue) and actin cytoskeleton (red) staining and normalised to control spheroid surface area (time = 0 h). Statistical significance was determined using a one-way ANOVA with Tukey’s post hoc test for pairwise comparison (***, *p* < 0.001; ****, *p* < 0.0001).

**Figure 5 cancers-17-03399-f005:**
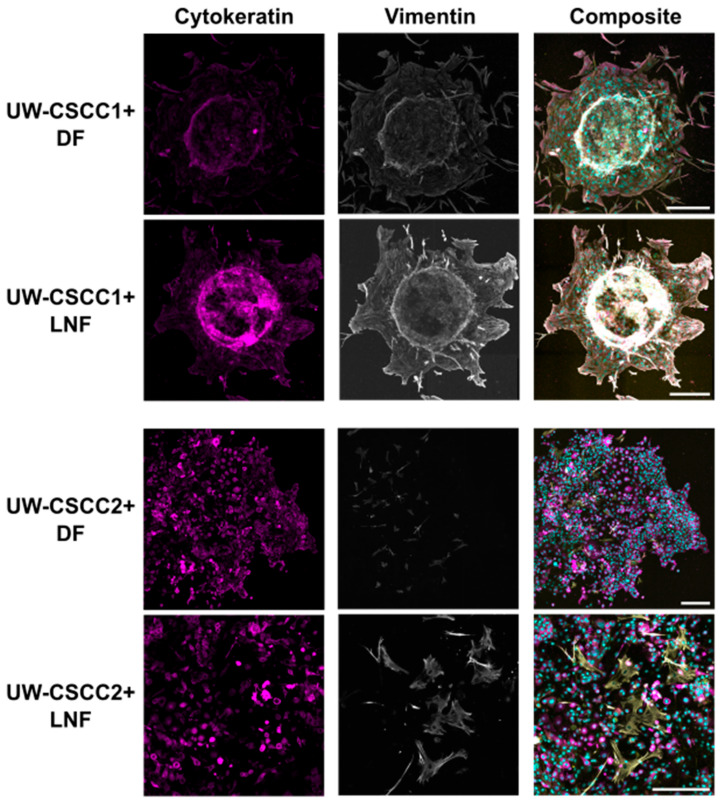
Pan-cytokeratin and vimentin expression in collagen-embedded invasive metastatic cSCC MCTS. UW-CSCC1 or UW-CSCC2 and either DF or LNF MCTS were embedded in collagen I. After 80 h of invasion, MCTS were stained and imaged for cytokeratin (magenta) and vimentin (grey). Nuclei (cyan) and actin cytoskeleton (yellow) were counterstained with Hoechst-33342 and ActinRed^TM^ 555, respectively. The images represent the summation of all slices of lightening processed images. Brightness and contrast were adjusted for presentation purposes. Scale bar represents 250 µm.

## Data Availability

Data are available from the corresponding authors upon reasonable request.

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
