# Peer review of "Spatial Organisation and Invasive Behaviour of Metastatic Cutaneous Squamous Cell Carcinoma-Derived Multicellular Spheroids Reflect Tumour Cell Phenotype"

_cancers, 2025, doi:10.3390/cancers17213399_

Round 1

Reviewer 1 Report

Comments and Suggestions for Authors

In this study, the authors employed three cell lines (UW-CSCC1-3) and two types of fibroblasts (DF and LNF) to investigate the formation of multicellular spheroids and the invasive properties of cancer cells. The formation of these spheroids and the related experimental analyses (e.g., cell staining) are technically challenging. The authors attempted to clarify the intrinsic nature of cancer cells as well as the possible in vivo roles of fibroblasts. While this line of investigation is indeed valuable, the presentation in the manuscript is at times difficult to follow and not entirely clear. I have several comments and questions for the authors’ consideration.

Major Comments:

  1. The study primarily focuses on UW-CSCC1-3 cells; however, most of the presented data are centered on UW-CSCC1 and UW-CSCC2. Since the intrinsic characteristics of UW-CSCC3 fall between those of the other two cell lines—and many subsequent results also reflect this intermediate behavior—its data are mostly shown in the supplementary material. I also noticed that the sampling times for UW-CSCC1-2 and UW-CSCC3 differ, likely because the experiments were performed at different periods, which makes it difficult to directly compare results within the same figure. For these reasons, I recommend moving all UW-CSCC3 data, including its introduction in Figure 1, to the supplementary material. This restructuring would improve the overall coherence and readability of the manuscript.
  2. In the Introduction (lines 65–82), the authors cited previous studies suggesting that reticular fibroblasts (LNF), which express α-SMA and vimentin, possess a stronger ability than dermal fibroblasts (DF) to enhance cancer cell metastasis. However, in the present results (Figure 4), LNFs do not appear to promote CSCC1 invasion, whereas DF unexpectedly increases the invasiveness of the otherwise less aggressive CSCC2. Although this issue is briefly acknowledged in the Discussion (lines 390–393), the explanation is not sufficiently in-depth and lacks the robustness expected, especially compared to how it was framed in the Introduction.
  3. The results presented in Figure 3 are not easy to interpret. For instance, two composite images are provided, yet their differences are not clearly explained. Moreover, in UW-CSCC1+LNF spheroids, the absence of yellow staining in the B1–B2 core region raises the question of whether fibroblasts are indeed absent from this area. In addition, it is unclear whether the cancer cells lack actin cytoskeleton expression. These staining results require more detailed clarification.
  4. Several images appear to be poorly cropped, making comparisons between groups look rough—for example, UW-CSCC2+DF in Figure 3 and parts of Figure 5. Furthermore, the scale bar fonts and line thicknesses should be standardized across all figures.
  5. The figure legends should specify the time points at which the images were acquired—for instance, Figure 2B and 2D (60 h?), Figure 4B and 4D (48 h and 18 h?), and Figure 5.
  6. Some data citations need to be improved. For example, in line 217, since the authors mention Western blot results, it would be more appropriate to also cite Supplementary Figure 4 in addition to Figure 1.
  7. In Supplementary Figure S6, the expression level of E-cadherin in the UNT group appears unusual and does not match the description provided in lines 321–323. I suggest replacing this figure.

Author Response

Comments 1: The study primarily focuses on UW-CSCC1-3 cells; however, most of the presented data are centered on UW-CSCC1 and UW-CSCC2. Since the intrinsic characteristics of UW-CSCC3 fall between those of the other two cell lines—and many subsequent results also reflect this intermediate behavior—its data are mostly shown in the supplementary material. I also noticed that the sampling times for UW-CSCC1-2 and UW-CSCC3 differ, likely because the experiments were performed at different periods, which makes it difficult to directly compare results within the same figure. For these reasons, I recommend moving all UW-CSCC3 data, including its introduction in Figure 1, to the supplementary material. This restructuring would improve the overall coherence and readability of the manuscript.

Response 1: We thank the reviewer for the comment and agree that these changes would improve the readability of the manuscript. As suggested, we have moved all UW-CSCC3 data to the supplementary material (Supplementary Figure S5). This specifically includes: Removing detail surrounding UW-CSCC3 from methods section; moving UW-CSCC3 data out of Figure 1 and into Supplementary Figure S5; Removal of text in Results section 3.1 describing this cell line. We have added additional panels to this updated Supplementary Figure S5 to include relevant details surrounding this cell line’s origin.

Comments 2: In the Introduction (lines 65–82), the authors cited previous studies suggesting that reticular fibroblasts (LNF), which express α-SMA and vimentin, possess a stronger ability than dermal fibroblasts (DF) to enhance cancer cell metastasis. However, in the present results (Figure 4), LNFs do not appear to promote CSCC1 invasion, whereas DF unexpectedly increases the invasiveness of the otherwise less aggressive CSCC2. Although this issue is briefly acknowledged in the Discussion (lines 390–393), the explanation is not sufficiently in-depth and lacks the robustness expected, especially compared to how it was framed in the Introduction.

Response 2: We did not mean that there are previous studies suggesting that LNFs (which are indeed reticular) possess a stronger ability to enhance cancer metastasis than dermal fibroblasts (DFs), which can be of either reticular or papillary origin. We cited papers saying that subtypes of dermal (not LNF) derived fibroblasts (reticular vs papillary) may impact tumour cell invasiveness dependent on the model used as there is no literature that we are aware of that has compared lymph node-derived fibroblasts to dermally-derived fibroblasts in models of cSCC invasiveness. As our study is the first to compare LNFs to DFs (regardless of subtype), we approached this study with no expectations to what extent these fibroblasts would differentially influence cSCC invasion. To avoid confusion for future readers we have made this point clearer by modifying (underlined part) the relevant sentence to say:

However, how lymph node-derived fibroblasts (LNFs) compared to dermally-derived fibroblasts (DFs) shape the metastatic niche in the affected lymph nodes, their influence on EMT, and their expression of CAF markers in the context of metastatic cSCC remains unclear.” (Lines 81-84 of revised manuscript).

Further, in referring to why the DFs but not LNFs increased the invasiveness of CSCC2 only, we have added the following sentence and new citations to lines 411-413 in the Discussion of the revised manuscript:

“This could be due to fact that the LNFs were derived from uninvolved lymph nodes whereas the DFs are transformed fibroblasts and more likely to behave as CAFS which are known to help drive tumor cell EMT and invasiveness.” 

Comments 3: The results presented in Figure 3 are not easy to interpret. For instance, two composite images are provided, yet their differences are not clearly explained. Moreover, in UW-CSCC1+LNF spheroids, the absence of yellow staining in the B1–B2 core region raises the question of whether fibroblasts are indeed absent from this area. In addition, it is unclear whether the cancer cells lack actin cytoskeleton expression. These staining results require more detailed clarification.

Response 3: We have added labels for each composite and clarified within the figure caption that they refer to the spheroids stained separately with cytokeratin or E-cadherin (due to antibody species reactivity). We are unsure as to what the reviewer refers to regarding the absence of yellow staining in the core region for UW-CSCC1+LNF spheroids – there is yellow staining in this region (indicative of fibroblasts) which aligns with Figure 2 and the presence of fibroblasts within this core. We have further explicitly clarified that the cancer cells do express actin in the cytoskeleton within the results text to avoid further confusion: “Metastatic cSCC cells were positively identified by staining for cancer cell-specific cytokeratin, a nucleus, and actin cytoskeleton” [Lines 292-295 of the revised manuscript] 

Comments 4: Several images appear to be poorly cropped, making comparisons between groups look rough—for example, UW-CSCC2+DF in Figure 3 and parts of Figure 5. Furthermore, the scale bar fonts and line thicknesses should be standardized across all figures.

Response 4: These figures have been revised as requested. The cropping has been adjusted so each image has the same aspect ratio to improve the presentation, and scale bar fonts and thicknesses have been standardised in each figure.

Comments 5: The figure legends should specify the time points at which the images were acquired—for instance, Figure 2B and 2D (60 h?), Figure 4B and 4D (48 h and 18 h?), and Figure 5.

Response 5: We have rectified this. Figure 2B,D caption now specifies 60h time point. Figure 4B,D caption now specifies invasion time points as 48h and 18h respectively. Figure 5 already states 80h timepoint for the assay, but we have now specified that this time point also applies to the imaging in this figure within the caption [line 377 of the revised manuscript].

Comments 6: Some data citations need to be improved. For example, in line 217, since the authors mention Western blot results, it would be more appropriate to also cite Supplementary Figure 4 in addition to Figure 1.

Response 6: We have addressed this and cited Supplementary Figure 4 in this line [now line 226 of the revised manuscript]

Comments 7: In Supplementary Figure S6, the expression level of E-cadherin in the UNT group appears unusual and does not match the description provided in lines 321–323. I suggest replacing this figure.

Response 7: We have replaced the Western blot and related densitometry with a repeated experiment in Supplementary Figure S6, where the UNT bands are clearer. This still aligns with the description in text, but should avoid any uncertainty associated with the smeared band included in the previous iteration.

Reviewer 2 Report

Comments and Suggestions for Authors

Reviewer Comments to the Authors

General Comments

This study investigates how different combinations of patient-derived metastatic cutaneous squamous cell carcinoma (cSCC) cell lines and fibroblasts (from skin or lymph nodes) affect the structure and invasion behavior of 3D multicellular tumor spheroids (MCTS). Using live-cell imaging, the researchers analyzed the spatial organization and expression of key markers related to epithelial-to-mesenchymal transition (EMT) and cancer-associated fibroblasts (CAFs). They found that while fibroblasts influence the spheroid structure, the invasive behavior is primarily determined by the EMT status of the cancer cells. The study supports the use of MCTS as relevant models for studying tumor-stroma interactions in metastatic cSCC.

Specific Comments

A. Abstract

This section is solid both in scientific content and iintent. I have the following suggestions to improve its quality:

  • The phrase “for the first time”. Please make sure that this claim is definitively justified.
  • Please explicitly mention the “Research hypothesis” in this section.
  • Some sentences are overly long and should be split and simplified.
  • Please reduce the passive voice whenever possible.

 B. Introduction

The introduction section as it stands nicely provides a strong scientific foundation and context for the study. The background is comprehensive. The flow of the topics is logical. I have the following suggestions to improve the quality of this section:

  • Some sentences are long and packed with multiple ideas. The authors should break them into shorter, to be easily digestible by the reader. For instance: “In this reticular fibroblast/cSCC co-culture model, reticular fibroblasts expressed CAF biomarkers α-smooth muscle actin (α-SMA) and vimentin, while cSCC cells expressed EMT markers N-cadherin and ZEB1.”
  • The introduction outlines aims of the study but no “research hypothesis”. Please add a clear “Research hypothesis”
  • The authors should mention the specific research questions.
  • Please provide brief comments on the concepts and acronyms (MCTS, CAFs, EMT, CAFs, and fibroblast subtypes) once introduced. This will make the section digestible to the readers who are not expert in the field of cancer biology.
  • Use linking phrases such as “To build upon these findings,” to improve cohesion between different ideas.
  • Please elaborate more on the issue of “multicellular tumour spheroids (MCTS) [14, 15]”

C. Materials and Methods

This section is well-written and scientifically sound. Also, I am happy with the supplementary materials which are appropriately referenced. I have the following suggestions to improve its clarity:

  • At the very beginning of this section, I would like to see a brief overview of the study design. The authors may be willing to add a flow-chart summarizing the experimental design.
  • What is the rational behind using these specific cell lines (UW-CSCC1 and UW-CSCC2)?
  • You mention the derivation of a new cell line (UW-CSCC3) but provide no details on its characterization. How do you confirm it's representative of cSCC? Was it validated the same way as UW-CSCC1/2?
  • Why was Katushka2S chosen specifically? You describe its properties, but you could briefly link this to the imaging strategy.
  • Please elaborate more on the following: Trypsinisation method for removing contaminants in LNF cultures is mentioned but not described (add time, concentration, and rationale for differential trypsinization). Also please elaborate on the Fluorescence-assisted cell sorting (FACS).

D. Results

This section is well-written and scientifically sound.

E. Discussion

Overall, this section falls short of expectations in both depth and structure. As it stands, it is brief and lacks the critical analysis of the results. Much of the content simply reiterates existing literature.

The discussion should begin with a concise summary of the key findings, clearly enumerated (e.g., i), ii), iii)) to provide the reader with a structured overview. Following this, the section should be reorganized under thematic subheadings, each dedicated to a major finding of the study. Within each subsection, relevant comparisons should be drawn to similar studies, including even those in non-cutaneous squamous cell carcinomas where appropriate. These comparisons should highlight similarities, differences, and potential reasons for any discrepancies in outcomes.

Furthermore, the molecular mechanisms underlying the observed phenomena should be discussed in greater detail to enhance the biological relevance of the findings. Toward the end of the discussion, the limitations of the current study should be clearly acknowledged, and specific questions for future research should be proposed to guide continued investigation in this area.

Author Response

Comments 1: Abstract. This section is solid both in scientific content and iintent. I have the following suggestions to improve its quality:

  • The phrase “for the first time”. Please make sure that this claim is definitively justified.
  • Please explicitly mention the “Research hypothesis” in this section.
  • Some sentences are overly long and should be split and simplified.
  • Please reduce the passive voice whenever possible.

Response 1: We thank the reviewers for their comment. We have removed the phrase “for the first time”. We have added an explicit statement of the research hypothesis: “We hypothesised that the interplay between different cSCC and fibroblast cell combinations would differentially influence spheroid formation and invasion” [lines 33-34 of the revised manuscript]. We have split long sentences within the “Results” subheading of the abstract [lines 38-40 of the revised manuscript]. Passive voice has been removed where possible – e.g., “Our aim was” to “We aimed”.

Comments 2: Introduction. The introduction section as it stands nicely provides a strong scientific foundation and context for the study. The background is comprehensive. The flow of the topics is logical. I have the following suggestions to improve the quality of this section:

  • Some sentences are long and packed with multiple ideas. The authors should break them into shorter, to be easily digestible by the reader. For instance: “In this reticular fibroblast/cSCC co-culture model, reticular fibroblasts expressed CAF biomarkers α-smooth muscle actin (α-SMA) and vimentin, while cSCC cells expressed EMT markers N-cadherin and ZEB1.”
  • The introduction outlines aims of the study but no “research hypothesis”. Please add a clear “Research hypothesis”
  • The authors should mention the specific research questions.
  • Please provide brief comments on the concepts and acronyms (MCTS, CAFs, EMT, CAFs, and fibroblast subtypes) once introduced. This will make the section digestible to the readers who are not expert in the field of cancer biology.
  • Use linking phrases such as “To build upon these findings,” to improve cohesion between different ideas.
  • Please elaborate more on the issue of “multicellular tumour spheroids (MCTS) [14, 15]

Response 2: We thank the reviewer for their detailed comments, and have addressed their suggestions as follows:

  • We have split the specified long sentence into shorter ideas that are easier to read. We have also split other longer sentences, including: “Characterisation of ligand-receptor interactions identified TGF-β as a central regulator of CAF activation. This was in line with previous findings supporting a role of TGF-β-activated CAFs in inducing EMT and facilitating invasion in cSCC” [line 65-67 of the revised manuscript]. 
  • We added an explicit research hypothesis in the final paragraph of the introduction, alluding to the research questions: “We hypothesised that the EMT status of the cancer cell and the origin of fibroblast used would differentially influence the MCTS phenotype” [lines 98-100 of the revised manuscript]
  • We have added a definition for EMT where it is first mentioned to improve clarity: “Epithelial-to-mesenchymal transition (EMT) is the dynamic process whereby epithelial cells lose their adhesive properties and acquire mesenchymal features associated with a more invasive phenotype.” [lines 57-59]. We have added new citations to accompany this statement. We have also added a description of CAFs, where they are first mentioned in the introduction: “a stromal cell type comprising a large proportion of the tumour microenvironment” and cited additional new references to accompany this statement [lines 62-63]

  • We are unsure of where the suggested type of linking sentence would be relevant, and feel the flow of the introduction is acceptable.

  • We have elaborated on MCTS where they are introduced: “MCTS are self-assembling three-dimensional spheroids containing multiple distinct cell types to replicate the cellular composition of the microenvironment.” [lines 88-90]

Comments 3: Methods. This section is well-written and scientifically sound. Also, I am happy with the supplementary materials which are appropriately referenced. I have the following suggestions to improve its clarity:

  • At the very beginning of this section, I would like to see a brief overview of the study design. The authors may be willing to add a flow-chart summarizing the experimental design.
  • What is the rational behind using these specific cell lines (UW-CSCC1 and UW-CSCC2)?
  • You mention the derivation of a new cell line (UW-CSCC3) but provide no details on its characterization. How do you confirm it's representative of cSCC? Was it validated the same way as UW-CSCC1/2?
  • Why was Katushka2S chosen specifically? You describe its properties, but you could briefly link this to the imaging strategy.
  • Please elaborate more on the following: Trypsinisation method for removing contaminants in LNF cultures is mentioned but not described (add time, concentration, and rationale for differential trypsinization). Also please elaborate on the Fluorescence-assisted cell sorting (FACS).

Response 3: We thank the reviewer for their suggestions and have addressed as following:

  • We feel that the concluding sentence of our Introduction sets the scene already and that the experimental design is straightforward enough that an introductory overview and/or flow chart is unnecessary.

  • These cell lines were selected as they are two of the few robustly characterised metastatic cSCC cell lines available. They also represent different ends of the EMT spectrum and were anticipated to therefore behave differently within MCTS to investigate our research question, which we had already described in detail in the opening section of the Results. Nevertheless, we have reworded the sentences first referring to these cell lines in Methods Section 2.1 as follows:

    The patient-derived cell lines UW-CSCC1 and UW-CSCC2, originating from lymph node metastases of head and neck cSCC, were utilised for the generation of MCTS and were routinely cultured as previously described [25]. These cell lines have been extensively characterised and exhibit distinctive genomic and phenotypic profiles in vitro, mirroring both the features of their tumour of origin and the heterogeneity observed between tumours.[25]”. [lines 106-110]

  • The cell line UW-CSCC3 was developed and validated as previously described – we have added this detail to the caption of Supplementary Figure S5 (please note that details surrounding UW-CSCC3 have been moved to supplementary material as suggested by Reviewer 1).

  • Katushka2S was selected as the far-red excitation and emission wavelength coupled with the exceptional brightness allow Katushka2S to perform even in challenging samples such as MCTS. We have now added this detail to the Methods section 2.3 [lines 136-137].

  • We have added details of this within Methods 2.2: “Other potentially contaminating cells were removed via differential trypsinisation, based on the rapid detachment of fibroblasts. Cultures were incubated with 0.05% trypsin-EDTA for 30 second intervals until cells visibly lifted.” [lines 124-127]

    With regards to FACS, it is unclear to us what more FACS related details the reviewer is requesting we add as we already included the instrument settings and gating strategy as required. We refer the reviewer to Supplementary Figure S3 for further details beyond the methods text.

Comments 4: Results. This section is well-written and scientifically sound.

Response 4: We thank the reviewer and note that Figures 1, 3, 4 and 5 have been improved based on feedback from Reviewer 1. 

Comments 5: Discussion. Overall, this section falls short of expectations in both depth and structure. As it stands, it is brief and lacks the critical analysis of the results. Much of the content simply reiterates existing literature. The discussion should begin with a concise summary of the key findings, clearly enumerated (e.g., i), ii), iii)) to provide the reader with a structured overview. Following this, the section should be reorganized under thematic subheadings, each dedicated to a major finding of the study. Within each subsection, relevant comparisons should be drawn to similar studies, including even those in non-cutaneous squamous cell carcinomas where appropriate. These comparisons should highlight similarities, differences, and potential reasons for any discrepancies in outcomes. Furthermore, the molecular mechanisms underlying the observed phenomena should be discussed in greater detail to enhance the biological relevance of the findings. Toward the end of the discussion, the limitations of the current study should be clearly acknowledged, and specific questions for future research should be proposed to guide continued investigation in this area.

Response 5: We thank the reviewer for providing detailed feedback. As requested, we have separated the discussion into clear, subtitled sections with the key findings. We have more concisely summarised key findings at the beginning (paragraph 1 of the discussion), and added more critical interpretation of the results with reference to the literature – this includes similarities, differences, and potential explanations for differences as requested. We have clearly acknowledged the limitations within.  

Round 2

Reviewer 1 Report

Comments and Suggestions for Authors

The author has basically addressed all of my  comments, and the presentation of the graphics has been greatly improved.

Reviewer 2 Report

Comments and Suggestions for Authors

None